# Clinicopathological Significances of Tumor–Stroma Ratio (TSR) in Colorectal Cancers: Prognostic Implication of TSR Compared to Hypoxia-Inducible Factor-1α Expression and Microvessel Density

**Guhyun Kang** [1],[†] 📴 **, Jung-Soo Pyo** [2],[†] 📴 **, Nae-Yu Kim** [3] **and Dong-Wook Kang** [4],[5],[*] 📴

[1] Department of Pathology, Daehang Hospital, Seoul 06699, Korea; guhyunkang@daum.net
[2] Department of Pathology, Uijeongbu Eulji University Hospital, Eulji University School of Medicine, Gyeonggi Province, Kyeonggi-Do 11759, Korea; jspyo@eulji.ac.kr
[3] Department of Internal Medicine, Daejeon Eulji University Hospital, Eulji University School of Medicine, Daejeon 35233, Korea; naeyu46@eulji.ac.kr
[4] Department of Pathology, Chungnam National University Sejong Hospital, 20 Bodeum 7-Ro, Sejong 30099, Korea
[5] Department of Pathology, Chungnam National University School of Medicine, 266 Munhwa Street, Daejeon 35015, Korea
[*] Correspondence: astro966@gmail.com; Tel.: +82-10-8561-9895
[†] Guhyun Kang and Jung-Soo Pyo have contributed equally to this study.

**Abstract:** The present study aimed to elucidate the clinicopathological significance and prognostic implications of tumor–stroma ratio (TSR) in colorectal cancers (CRCs). TSRs were investigated in 266 human CRC specimens. The correlations between TSR and clinicopathological characteristics and survival were evaluated. The hypoxia-inducible factor-1α (HIF-1α) immunohistochemical expression of tumor cells and microvessel density (MVD) of stroma were compared between stroma-low and stroma-high subgroups. Results: Stroma-low was found in 185 of 266 CRCs (69.5%). Stroma-low was significantly correlated with less frequent vascular and perineural invasion and distant metastasis than stroma-high. HIF-1α of tumor cells was more highly expressed in the stroma-high subgroup than in the stroma-low subgroup. In addition, MVD was significantly higher in the stroma-high subgroup compared to the stroma-low subgroup. The stroma-low rate was increased considerably in CRCs with a mucinous component and decreased in CRCs with a micropapillary component. There were significant correlations between stroma-low and better overall and recurrence-free survivals. Similar to the literature, we observed that stroma-low was significantly correlated with favorable tumor behaviors and better survival. The microscopic examination of TSR can be useful for predicting the prognosis of CRC patients.

**Keywords:** tumor–stroma ratio; colorectal cancers; clinicopathological significance; hypoxia-inducible factor-1; microvessel density

## 1. Introduction

Epithelial malignant tumors initially invade the basement membrane and sequentially progress to the stroma. The interaction between the tumor cells and stroma, including the extracellular matrix, is important in the tumor progression of many cancers [1]. On the other hand, tumor progression can be suppressed by peritumoral stromal components [2]. The extracellular matrix of the stroma provides a dynamic network in tumor cells and affects tumor behavior [1]. As the tumor size varies, the amount of stroma can also vary. The interface between tumor cells and stroma may be necessary for interpreting the role of the stroma. Colorectal cancer (CRC) is one of the common cancers worldwide [3]. In normal colorectal tissue and CRC, lymphocytic infiltrations are frequently identified [4–7]. In addition, the prognostic role of tumor-infiltrating lymphocytes and the immunoscore

has been reported in CRC [7,8]. These results suggest the crucial roles of intratumoral stroma in CRC. The tumor–stroma ratio (TSR) has been introduced as a simple parameter for evaluating the intratumoral stroma.

Although previous studies reported the prognostic impact of TSR in CRC, the results are controversial [9–14]. Zengin et al. reported that tumors with a high proportion of stroma (>50% stroma, defined as stroma-high) were significantly correlated with worse survival, whereas tumors with an abundant carcinoma component (≤50% stroma, stroma-low) were associated with a better prognosis [14]. In addition, there was no significant correlation between TSR and survival [13]. A detailed analysis based on histologic subtypes has not been performed. Besides, the evaluation criteria for TSR are not fully elucidated. Basically, malignant tumors are evaluated for deciding therapy and predicting a patient's prognosis through the tumor node metastatic (TNM) classification [15]. Because TSR can easily be assessed through routine pathologic examinations, it is important to elucidate the evaluation criteria for TSR in CRC.

In the present study, we investigated the TSR of CRCs and evaluated the clinicopathological significance and prognostic implications of TSR. Also, we performed a subgroup analysis based on cutoffs of TSR and histologic subtypes. Angiogenesis is a multi-step process involving many angiogenesis-related molecules, such as hypoxia-inducible factor-1$\alpha$ (HIF-1$\alpha$). Angiogenesis is regulated by multiple factors, many of which may individually predict microvessel density (MVD). Besides, since the tumor progression requires angiogenesis in the stroma, we compared the angiogenesis between stroma-low and stroma-high subgroups with the HIF-1$\alpha$ expression of tumor cells and MVD of the stroma.

## 2. Materials and Methods

### 2.1. Cases Selection and Specimens

The clinical histories of 266 patients who underwent surgical resection of CRC at the Eulji University Medical Center between 1 January 2001 and 31 December 2010 were analyzed. The specimens obtained from the patients who received preoperative neoadjuvant therapy were excluded from this study. All cases were histologically confirmed as primary colorectal adenocarcinoma and re-examined by two independent pathologists (G.H.K. and J.S.P.). We reviewed the medical charts, pathological records, and glass slides in order to assess the clinicopathological characteristics of each individual, such as TSR; age; sex; tumor size; tumor location; tumor differentiation; vascular, lymphatic, and perineural invasion; depth of tumor; lymph node metastasis; distant metastasis; and pathological TNM stages. These cases were evaluated according to the eighth edition of the American Joint Cancer Committee TNM classification [15]. Cases with R0 and R1 resections were 264 and 2 cases, respectively. Histological subtypes included adenocarcinoma and mucinous adenocarcinoma. Some cases had micropapillary or signet-ring cell components. Clinical outcomes were followed from surgery date to either the date of death or five years, resulting in a follow-up period ranging from 0 to 60 months. This protocol was reviewed and approved by the Institutional Review Board of the Eulji University Hospital (Approval No. EMC 2020-09-011).

### 2.2. Evaluation of Tumor–Stroma Ratio

We prepared glass slides with hematoxylin and eosin (H&E) staining from the formalin-fixed paraffin-embedded section to evaluate the TSR. Because the amount of stroma can be different within the tumor, all H&E slides were screened and evaluated for TSR. TSR is defined as the proportion of the tumor cells relative to the surrounding stroma in the overall tumor bed [16]. The assessment of TSR was conducted using scoring percentages in 10% increments (10%, 20%, 30%, etc.). The mucinous component was evaluated as a tumor component in assessments of TSR. We divided TSR into high and low subgroups, using a cutoff of 50%. Conflicting cases were re-examined, and consensuses between three pathologists (Kang G., Pyo J.S., and Kang D.W.) were reached.

## 2.3. Tissue Microarrays and Immunohistochemistry

For immunohistochemical staining, five array blocks containing the 266 resected CRC tissue cores obtained from all patients were prepared. Briefly, tissue cores (2 mm in diameter) were retrieved from each paraffin-embedded CRC tissue sample (donor blocks) and arranged in recipient paraffin blocks using a trephine apparatus, as described previously [17]. The staining of the different intratumoral areas in these tissue array blocks showed excellent agreement among observers. A core was chosen from each adequate case for analysis. An adequate case was defined as a tumor occupying more than 10% of the core area. Each block contained internal controls consisting of non-neoplastic colon tissue.

For immunohistochemistry, sections with 4 μm thickness were cut from each tissue array block, deparaffinized, and dehydrated. Immunohistochemical staining was conducted following the compact polymer method using a VENTANA benchmark XT autostainer (Ventana Medical Systems, Inc., Oro Valley, AZ, USA) and visualized by treatment with an OPTIVIEW universal 3,3′-diaminobenzidine kit (Ventana Medical Systems, Inc.). Sections were then incubated with an anti-cluster differentiation 34 (anti-CD34; Cell Marque, Darmstadt, Germany) and anti-HIF-1α (Novus Biologicals, Littleton, CO, USA). To confirm the reaction specificity of the antibodies, a negative control stain without a primary antibody was performed. All immunostained sections were lightly counterstained with Mayer's hematoxylin.

HIF-1α immunohistochemical expression was evaluated in the nuclei of tumor cells. The intensity of each protein's expression in the immunohistochemically stained samples was scored using a scale from 0 to 3 (0 = negative; 1 = weak; 2 = moderate; 3 = strong). In addition, the percentage of positively stained cells was categorized on the basis of a scoring system from 0 to 4 (1 = 0–25%; 2 = 26–50%; 3 = 51–75%; 4 = 76–100%). An immunoreactive score (IRS) was calculated by multiplying the staining intensity score by the percentage of positively stained cells [18]. Staining patterns were classified as negative (IRS: 0–4) or positive (IRS: 6–12).

## 2.4. Evaluation of Microvessel Density

MVD in the stroma was evaluated by immunohistochemistry for CD34, as described previously [19]. The three most highly vascularized areas in areas of tumor sections were selected under ×100 magnification, and photographs of CD34-immunopositive microvessels in tumor sections were taken under ×400 magnification using light microscopy. The cross-sectional areas of CD34-immunopositive structures (i.e., MVD) were quantified by capturing images, converting them to grayscale, and analyzing CD34-stained areas using the National Institutes of Health (NIH) Image Analysis software (version 1.62; National Institutes of Health, Bethesda, MD, USA) after setting one consistent intensity threshold for all slides. Then, CD34-positive areas were expressed as pixels squared per high-power field. According to the median microvessel count of all patients, samples were classified into high-MVD and low-MVD groups. Conflicting cases were re-examined, and consensuses were reached.

## 2.5. Statistical Analyses

Statistical analyses were performed using SPSS version 22.0 software (IBM Co., Chicago, IL, USA). A $\chi^2$ test determined the significance of the correlation between the TSR and the patients' clinicopathological characteristics. The comparisons between TSR and age, tumor size, or metastatic lymph node ratio were analyzed using a two-tailed Student's *t*-test. Survival curves were estimated using the Kaplan–Meier product-limit method, and differences between the survival curves were determined to be significant on the basis of the log-rank test. Results were considered statistically significant at $p < 0.05$.

## 3. Results

### 3.1. Correlation between Tumor–Stroma Ratio and Clinicopathological Characteristics

To evaluate TSR's clinicopathological significance and prognostic implications, we divided the TSR into stroma-high and stroma-low subgroups. Figure 1 showed representative images of stroma-low (Figure 1A,C) and stroma-high (Figure 1B,D), respectively. Stroma-low was found in 185 of 266 CRCs (69.5%). Stroma-low was significantly correlated with less frequent vascular and perineural invasion and distant metastasis (Table 1). There was no significant correlation between TSR and other clinicopathological characteristics, such as tumor size, tumor differentiation, and pT stage. In comparing TSR and histologic subtypes, stroma-low was significantly correlated with the mucinous component ($p = 0.034$; Table 2). However, CRCs with a micropapillary component were significantly associated with stroma-high.

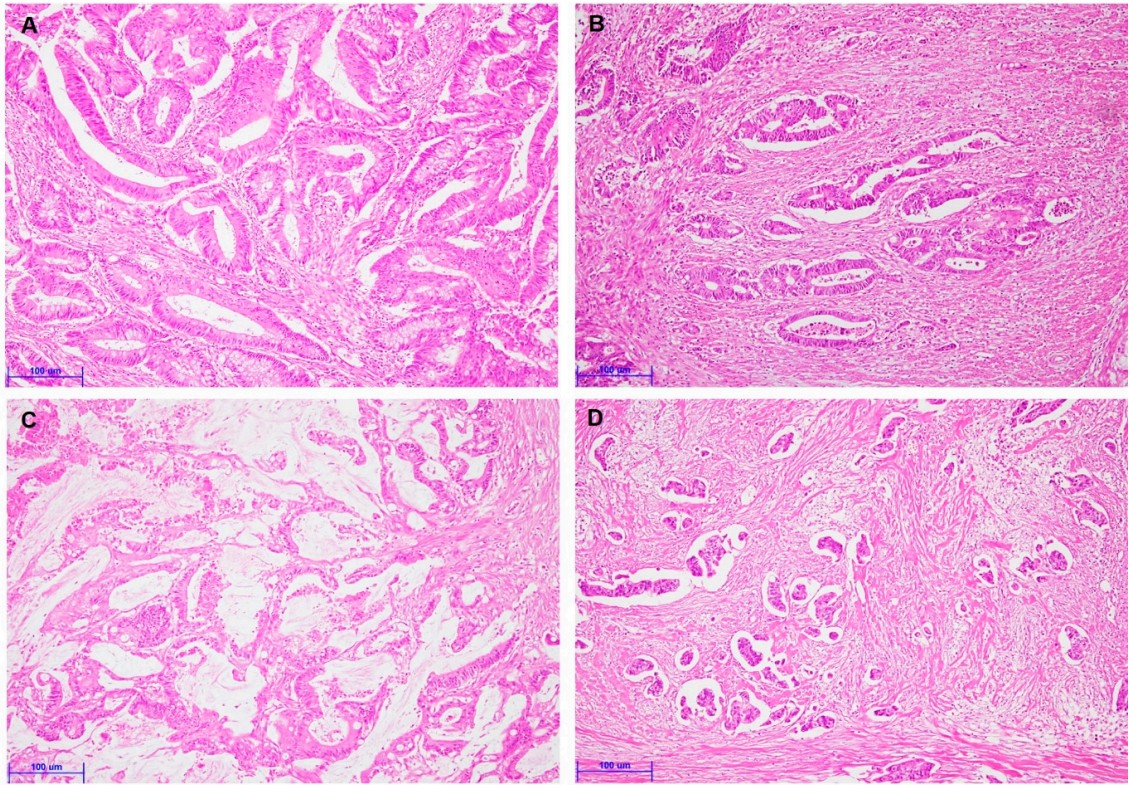

**Figure 1.** Representative images are showing colorectal cancers with various tumor–stroma ratios (**A–D**). (**A**) Colorectal cancer with stroma-low (×200). (**B**) Colorectal cancer with stroma-high (×200). (**C**) Colorectal cancer with the mucinous component (×200). (**D**) Colorectal cancer with a micropapillary component (×200). (Scale bar = 100 μm).

**Table 1.** The correlation between the tumor–stroma ratio and clinicopathological parameters in colorectal cancers.

| | Stroma | | *p*-Value |
|---|---|---|---|
| | **Low** | **High** | |
| Total (*n* = 266) | 185 (69.5) | 81 (30.5) | |
| Age (years ± SD) | 63.88 ± 12.97 | 62.91 ± 12.84 | 0.575 |
| Sex | | | |
| Male | 92 (49.7) | 43 (53.1) | 0.614 |
| Female | 93 (50.3) | 38 (46.9) | |
| Tumor size | | | |
| ≤5 cm | 72 (38.9) | 34 (42.0) | 0.639 |
| >5 cm | 113 (61.1) | 47 (58.0) | |

**Table 1.** *Cont.*

| | Stroma | | p-Value |
|---|---|---|---|
| | **Low** | **High** | |
| Tumor size (cm ± SD) | 5.58 ± 2.17 | 5.18 ± 1.83 | 0.142 |
| Location of tumor | | | |
| Right colon | 95 (51.4) | 33 (40.7) | |
| Left colon | 54 (29.2) | 29 (35.8) | 0.280 |
| Rectum | 36 (19.5) | 19 (23.5) | |
| Tumor differentiation | | | |
| Well | 15 (8.1) | 2 (2.5) | |
| Moderate | 130 (70.3) | 64 (79.0) | 0.161 |
| Poorly | 40 (21.6) | 15 (18.5) | |
| Vascular invasion | | | |
| Present | 10 (5.4) | 14 (17.3) | **0.002** |
| Absent | 175 (94.6) | 67 (82.7) | |
| Lymphatic invasion | | | |
| Present | 43 (23.2) | 27 (33.3) | 0.085 |
| Absent | 142 (76.8) | 54 (66.7) | |
| Perineural invasion | | | |
| Present | 19 (10.3) | 25 (30.9) | **<0.001** |
| Absent | 166 (89.7) | 56 (69.1) | |
| pT stage | | | |
| pT1-2 | 33 (17.8) | 8 (9.9) | 0.098 |
| pT3-4 | 152 (82.2) | 73 (90.1) | |
| Lymph node metastasis | | | |
| Present | 98 (53.0) | 48 (59.3) | 0.343 |
| Absent | 87 (47.0) | 33 (40.7) | |
| Distant metastasis | | | |
| Present | 15 (8.1) | 14 (17.3) | **0.027** |
| Absent | 170 (91.9) | 67 (82.7) | |
| pTNM stage | | | |
| I | 25 (13.5) | 7 (8.6) | |
| II | 59 (31.9) | 24 (29.6) | 0.280 |
| III | 86 (46.5) | 36 (44.4) | |
| IV | 15 (8.1) | 14 (17.3) | |

Numbers in parentheses represent percentage; SD, standard deviation; pT stage, pathologic tumor stage; pTNM stage, pathologic tumor node metastatic stage; $p < 0.05$ are highlighted in bold.

**Table 2.** The correlation between the tumor–stroma ratio and tumor subtypes in colorectal cancers.

| | Stroma | | p-Value |
|---|---|---|---|
| | **Low** | **High** | |
| Mucinous component | | | |
| Present | 38 (20.5) | 8 (9.9) | **0.034** |
| Absent | 147 (79.5) | 73 (90.1) | |
| Pure micropapillary component | | | |
| Present | 25 (13.5) | 24 (29.6) | **0.002** |
| Absent | 160 (86.5) | 57 (70.4) | |
| Distribution of micropapillary component | | | **0.048** |
| ≥5% | 12 (25.5) | 13 (48.1) | |
| <5% or negative | 35 (74.5) | 14 (51.9) | |

Numbers in parentheses represent percentage; $p < 0.05$ are highlighted in bold.

### 3.2. Correlation between Tumor–Stroma Ratio and Clinicopathological Characteristics

Next, we analyzed the correlation between TSR and angiogenesis markers due to its association with vascular invasion. Figure 2 showed representative images for the immunohistochemistry for HIF-1α and CD34. HIF-1α immunohistochemical expression of tumor cells was significantly higher in CRCs with stroma-high than those with stroma-low ($p = 0.035$). Moreover, MVD was considerably higher in the stroma-high subgroup than the stroma-low subgroup ($p = 0.008$). Table 3 shows the correlation between the various elements of TSR and CRCs. However, there was no significant correlation between HIF-1α and MVD ($p = 0.108$).

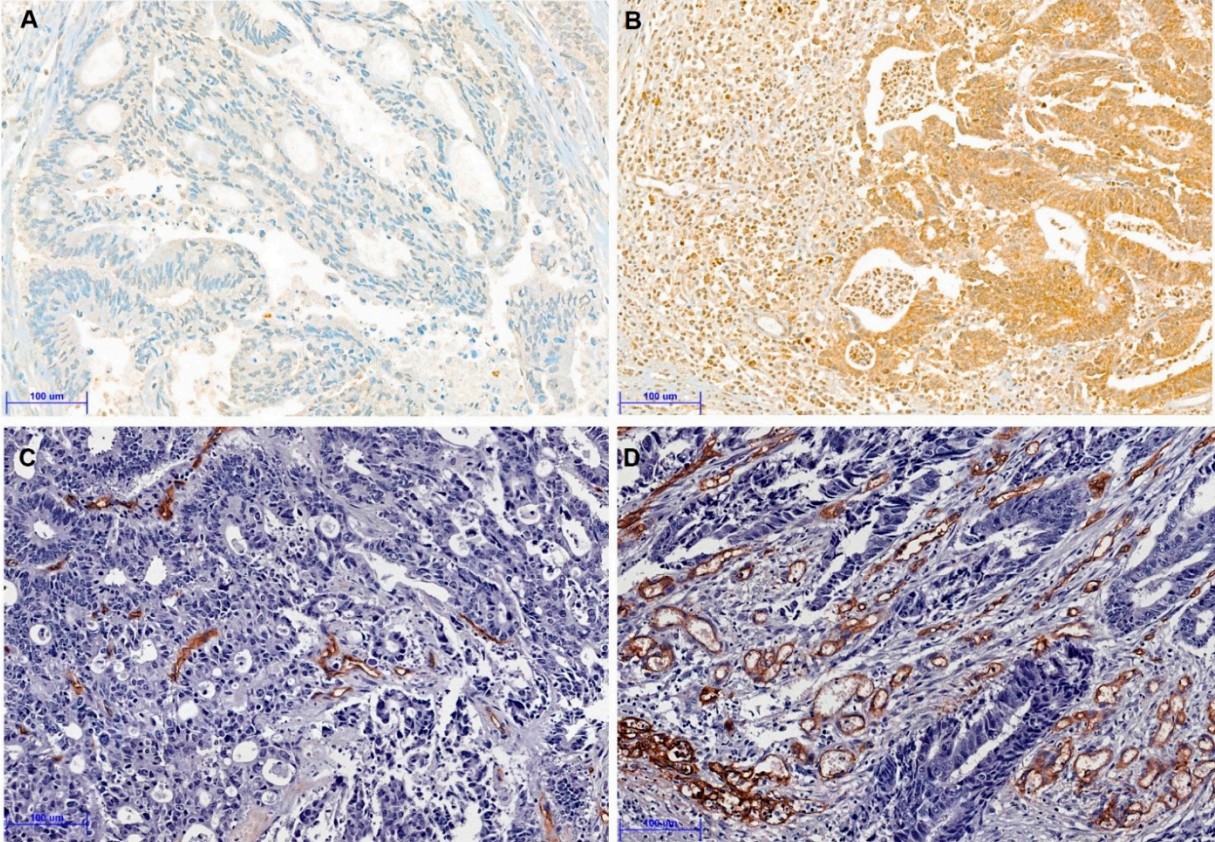

**Figure 2.** Representative immunohistochemical images for hypoxia-inducible factor-1α (HIF-1α) and CD34 (**A–D**). (**A**) Negative expression of HIF-1α in colorectal cancer with stroma-low (×200). (**B**) Positive expression of HIF-1α in colorectal cancer with stroma-high (×200). (**C**) Low CD34 expression of microvessel density (MVD) in colorectal cancer with stroma-low (×200). (**D**) High CD34 expression of MVD in colorectal cancer with stroma-high (×200). (Scale bar = 100 μm).

**Table 3.** The correlation between tumor–stroma ratio and various factors in colorectal cancers.

| | Stroma | | *p*-Value |
|---|---|---|---|
| | **Low** | **High** | |
| HIF-1α * | | | |
| Positive | 54 (29.2) | 34 (42.5) | **0.035** |
| Negative | 131 (70.8) | 46 (57.5) | |
| Microvessel density ** | | | |
| High | 82 (44.8) | 50 (62.5) | **0.008** |
| Low | 101 (55.2) | 30 (37.5) | |

HIF-1α, hypoxia-inducible factor-1α; *, One case lost the tissue for HIF-1α; Numbers in parentheses represent percentage; $p < 0.05$ are highlighted in bold; **, Three cases lost the tissue for CD34 immunohistochemistry.

### 3.3. Correlation between Tumor–Stroma Ratio and Survival

The prognostic impact of TSR was evaluated in CRCs. In the 50% cutoff subgroup, stroma-low was significantly correlated with better overall survival (OS) and recurrence-free survival (RFS) ($p < 0.001$ and $p < 0.001$, respectively; Figure 3). In the 30% cutoff subgroup, there was a significant correlation between stroma-low and better OS and RFS ($p = 0.005$ and $p = 0.008$, respectively; data not shown). However, in the 70% cutoff subgroup, there was no significant correlation between TSR and OS and RFS ($p = 0.0.89$ and $p = 0.133$, respectively).

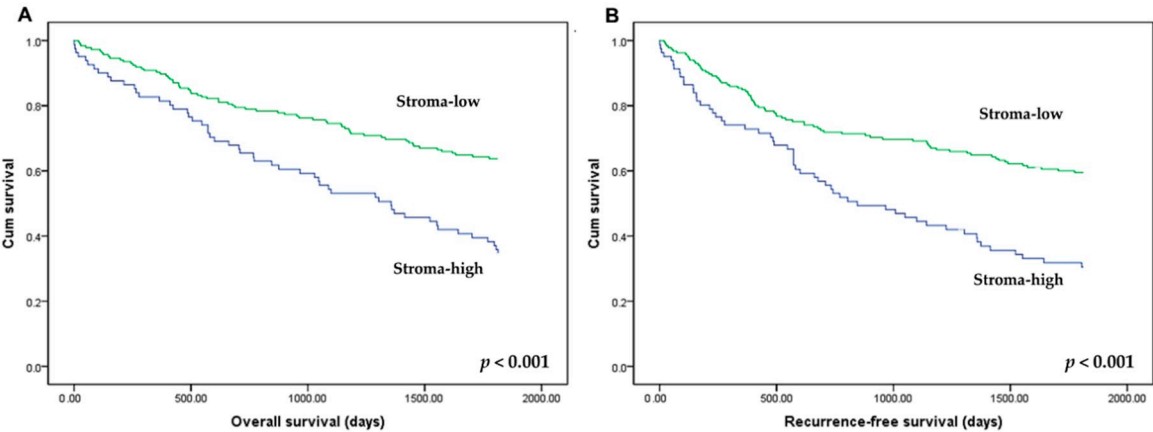

**Figure 3.** Kaplan–Meier curves for overall survival (**A**) and recurrence-free survival (**B**) according to the tumor–stroma ratio (TSR). Patients with stroma-high (low-TSR: ≤50% of tumor cells relative to stroma; blue line) and stroma-low (high-TSR: >50% of tumor cells relative to stroma; green line) showed overall and recurrence-free survivals.

There was no significant correlation between HIF-1α expression and OS and RFS ($p = 0.293$ and $p = 0.407$, respectively). In addition, there was no significant difference of OS and RFS between subgroups with high and low MVD ($p = 0.150$ and $p = 0.178$, respectively; Figure 4).

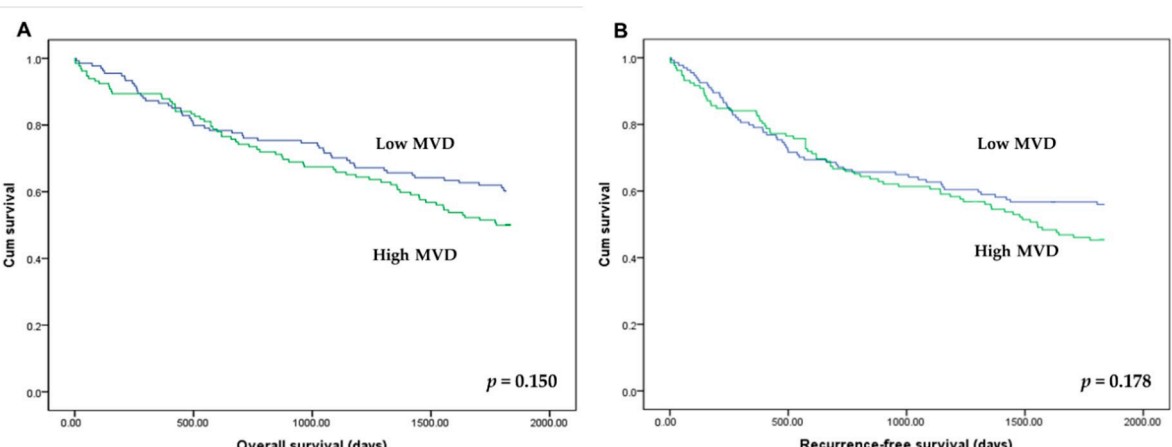

**Figure 4.** Kaplan–Meier curves for overall survival (**A**) and recurrence-free survival (**B**) according to the microvessel density (MVD). Patients with high (green line) and low MVD (blue line) showed overall and recurrence-free survivals.

### 4. Discussion

The present study aimed to elucidate the clinicopathological significance and prognostic implications of TSR in CRC. In addition, the prognostic impact of TSR was determined according to evaluation criteria for stroma-low. We found that stroma-low was significantly correlated with less frequent vascular invasion and lower MVD than stroma-high. The present study is the first, to the best of our knowledge, to elucidate (1) the correlation

between TSR and vascular invasion and MVD, and (2) the correlation between TSR and histologic subtypes in CRC.

TSR is defined as the proportion of the cancer cells relative to the surrounding stroma of the tumor [16]. In the tumor microenvironment concept, intratumoral stroma can be important in tumor growth through interactions with stroma. The assessment of intra- and peritumoral stroma can be critical in the microscopic examination. Tumors with a high percentage of stroma (stroma-high) may have a large contact area between tumor cells and stroma. This point can be considered for the active interaction between tumor cells and stroma. Previous studies reported the prognostic impact of TSR in CRCs. However, the prognostic implications of TSR differ among studies in CRCs [2,9,13,14,20,21]. Our results showed that stroma-low was significantly correlated with better OS and RFS in CRCs. However, Scheer et al. and Vogelaar et al. reported no significant correlation between TSR and survival in CRCs [2,13]. Discrepant results can be caused by various factors, including different study populations, evaluation criteria, and methods.

We evaluated TSR using scoring percentages in 10% increments (10%, 20%, 30%, etc.). In the present study, additional analysis on the basis of 30% and 70% cutoffs was performed. In the subgroup analysis, according to evaluation criteria for stroma-low, a 30% cutoff but not a 70% cutoff had a prognostic implication in comparisons between TSR and OS and RFS. According to our results, lower cutoffs, such as 30% or 50%, are more effective than higher criteria in predicting the patient's prognosis. In a previous study, Scheer et al. evaluated the prognostic impact of TSR by dividing their study sample into low ($\leq$30%), intermediate (40%, 50%, and 60%), and high ($\geq$70%) [13]. There was no significant difference in survival between the intermediate-TSR and low-TSR (stroma-high) subgroups. In our study, patients were classified into four subgroups based on the following cutoffs: TSR $\leq$ 30%, 30–50%, 50–70%, and $\geq$70%. Patients with TSR $\leq$ 30% and 30–50% had similar survival rates. There was no significant difference in survival rates between patients with TSR 50–70% and $\geq$70%. According to these results, a 50% cutoff may be appropriate to distinguish stroma-low.

Histologic subtypes of CRCs include micropapillary, mucinous, and medullary carcinomas, etc. [22]. The current study compared the TSR in CRCs with micropapillary and mucinous components. In our results, the rates of stroma-low differed among histologic subtypes. CRCs with a mucinous component showed a higher rate of stroma-low. However, CRCs with a micropapillary component were significantly correlated with stroma-high. This result suggested that TSR may be associated with the histologic characteristics of each subtype. The rule of assessment for TSR of the mucinous component has not been elucidated. In the present study, the mucinous component was evaluated as a tumor component in assessments of TSR. In a previous study, van Pelt GW et al. described that mucus, but not a mucinous component, should be ignored for scoring [23]. However, Huijbers et al. reported that the mucinous component was included as the stromal area [24]. If the entire tumor is composed of a mucinous component, the TSR is zero. This means that the tumor component is not present in tumors with TSR 0%. Moreover, mucinous components are included in mucin and floating tumor cells. The mucinous carcinoma is defined as a carcinoma with a greater than 50% mucinous component. Although the mucus without tumor cells is excluded, there was no specific reference for the interpretation of the mucinous component. Detailed information on the correlation between TSR and CRC with a mucinous component could not be obtained from previous studies [22]. In addition, there was no explanation for the TSR assessment of the mucinous component and mucin pool when using an automatic deep learning algorithm [25]. We previously reported the poor prognosis of CRCs with a micropapillary component [26]. In our previous study, CRCs with a micropapillary feature were significantly correlated with vascular invasion. Further studies will be needed to determine the clinicopathological implication of intratumoral stroma in CRC with a micropapillary component.

According to our results, TSR was significantly correlated with vascular and perineural invasion and distant metastasis. In addition, stroma-low was significantly correlated with

better survival in CRC. However, TSR was not associated with the pT stage and lymph node metastasis. Mesker et al. reported that TSR was useful in predicting prognosis for stage I–III colon cancers [10]. Interestingly, in a patient with the pT4 stage, there was no significant correlation between TSR and OS and RFS ($p = 0.812$ and $p = 0.915$, respectively). However, there was a significant correlation between TSR and survival rate in the pT1–3 stages (OS, $p < 0.001$; RFS, $p < 0.001$). In addition, regardless of lymph node metastasis, stroma-low was significantly correlated with better OS and RFS. A detailed examination of vascular and perineural invasions in stroma-high tumors is needed.

As a result, Huijbers et al. reported that a correlation between stroma-low and vascular invasion was found [19]. However, the relationship between the amount of stroma and vascular invasion is not fully understood. We investigated the MVD in the intratumoral stroma and the correlation between TSR and MVD in the present study. MVD was significantly higher in patients with stroma-high than in those with stroma-low. Also, the HIF-1$\alpha$ expression of tumor cells was significantly higher in the stroma-high subgroup than in the stroma-low subgroup. Additional subgroup analysis between high MVD and survival according to the TSR was added in the Supplementary Table (Table S1). Pancreatic cancers with high stroma demonstrated high vessel density [27]. Biologically, tumors with dense collagen content have good vascularization and are relatively well different [27]. However, evidence could not be found for the correlation between HIF-1$\alpha$ expression and the amount of stroma in colorectal cancers. The correlation between intratumoral stroma and hypoxic status is not clear. However, CRCs with less intratumoral stroma have low-MVD and may be susceptible to hypoxia.

## 5. Conclusions

In conclusion, stroma-low was significantly correlated with favorable tumor behaviors, including less frequent vascular and perineural invasion and distant metastasis. CRCs with stroma-low had better survival compared to those with stroma-high. In addition, a 50% cutoff may be appropriate for distinguishing stroma-low in predicting the patient's prognosis. In routine microscopic examination, TSR can be useful for predicting the prognosis of CRC patients.

**Supplementary Materials:** The following are available online at https://www.mdpi.com/1718-7729/28/2/125/s1, Table S1: The correlation between high microvessel density and overall and recurrence-free survivals in various subgroups according to the tumor stroma ratio.

**Author Contributions:** Conceptualization, G.K.; methodology, G.K. and J.-S.P.; software, J.-S.P.; validation, G.K. and J.-S.P.; formal analysis, J.-S.P.; data curation, N.-Y.K.; writing—original draft preparation, G.K. and J.-S.P.; funding acquisition, D.-W.K.; writing—review and editing, D.-W.K. All authors have read and agreed to the published version of the manuscript.

**Funding:** This work was supported by the research fund of Chungnam National University Hospital (2020-1679-01).

**Institutional Review Board Statement:** This study was conducted according to the guidelines of the Declaration of Helsinki, and approved by the Institutional Review Board of the Eulji University Hospital (Approval No. EMC 2020-09-011).

**Informed Consent Statement:** Not applicable.

**Data Availability Statement:** No new data were created or analyzed in this study. Data sharing is not applicable to this article.

**Acknowledgments:** We appreciated Joo Heon Kim (Department of Pathology, Eulji University Hospital, Eulji University School of Medicine) for the intellectual discussion and immunohistochemistry.

**Conflicts of Interest:** The authors declare that they have no potential conflicts of interest.

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
