# Peer review of "Clinicopathological Significances of Tumor–Stroma Ratio (TSR) in Colorectal Cancers: Prognostic Implication of TSR Compared to Hypoxia-Inducible Factor-1α Expression and Microvessel Density"

_curroncol, doi:10.3390/curroncol28020125_

Round 1
Reviewer 1 Report
In the current study, Kang et al. studied the clinicopathological significance and prognostic implications of tumor–stroma ratio (TSR) in colorectal cancers (CRCs). They also examined the co-relation among TSR, the expression of hypoxia-inducible factor-1alpha (HIF-1alpha), and microvessel density (MVD). Overall, the study was well-performed; however, there are several concerns to be addressed to properly interpret the data. Specific comments are as follows.
Major concerns
- It is unclear why HIF-1alpha and MVD were chosen as parameters to be compared with TSR? Are there any references?
- Line 86: What does the most representative cross-section mean?
- Figure 2: Representative pictures of both stroma-low and stoma-high cases should be presented.
- It seems likely that higher HIF-1alpha expression is correlated with higher MVD. Is this true? If so, discussion should be provided because it seems unusual.
- I’m curious to know why HIF-1alpha expression is higher in the stroma-high group. HIF-1alpha expression itself is an independent prognostic factor in CRC patients.
Minor concern
Scale bars should be included in micrographs.
Author Response
We tried to address the points raised by the reviewers as best as we can.
Please see the attachment.
In the current study, Kang et al. studied the clinicopathological significance and prognostic implications of tumor–stroma ratio (TSR) in colorectal cancers (CRCs). They also examined the co-relation among TSR, the expression of hypoxia-inducible factor-1alpha (HIF-1alpha), and microvessel density (MVD). Overall, the study was well-performed; however, there are several concerns to be addressed to properly interpret the data. Specific comments are as follows.
Major concerns
It is unclear why HIF-1alpha and MVD were chosen as parameters to be compared with TSR? Are there any references?
Response:
Angiogenesis is defined as the proliferation of blood vessels penetrating the tumor for the supply of nutrients and oxygen. In malignant tumors, angiogenesis is a requisite for continued tumor growth as well as for metastasis.
Angiogenesis is only occurred in the stroma.
Therefore, the impact of the amount of stroma can be important in evaluating tumor angiogenesis.
Tumor angiogenesis can be investigated through the measurement of microvessel density or microvessel area, or through the quantification of angiogenic molecules or angiogenic receptors in tumor tissues. Angiogenesis is a multi-step process involving many angiogenesis related molecules, such as HIF-1α. Angiogenesis is regulated by multiple factors, many of which may individually predict microvessel density.
As a recommendation, we added the explanation in the introduction as below:
Angiogenesis is a multi-step process involving many angiogenesis related molecules, such as hypoxia-inducible factor-1α (HIF-1α). Angiogenesis is regulated by multiple factors, many of which may individually predict microvessel density (MVD).
Line 86: What does the most representative cross-section mean?
Response:
We sectioned the entire tumor and grossly examined. Next, the representative sections were processing and microscopically examined.
The most representative cross-section is a section that represents the entire tumor among several cross-sections.
We corrected the comment in the revised manuscript as below:
Because the amount of stroma can be different within the tumor, all H&E slides were screened and selected as the most representative cross-section to evaluated TSR.
Figure 2: Representative pictures of both stroma-low and stroma-high cases should be presented.
Response:
We showed the representative pictures in Figure 1. Figure A and B represent colorectal cancers with stroma-low and stroma-high, respectively.
It seems likely that higher HIF-1alpha expression is correlated with higher MVD. Is this true? If so, discussion should be provided because it seems unusual.
Response:
We performed the additional analysis for the correlation between HIF-1α and MVD. In our study, there was no significant correlation between HIF-1α and MVD (p = 0.108).
We added the result in the revised manuscript.
However, there was no significant correlation between HIF-1α and MVD (p = 0.108).
I’m curious to know why HIF-1alpha expression is higher in the stroma-high group. HIF-1alpha expression itself is an independent prognostic factor in CRC patients.
Response:
We performed the additional analysis for the prognostic implication of HIF-1α expression. In our study, there was no significant correlation between HIF-1α expression and overall and recurrence-free survivals (p = 0.293 and p = 0.407, respectively).
In pancreatic cancer, a stroma-high phenotype demonstrated high vessel density and high collagen fraction (Klaassen et al., 2020). However, it could not be found the evidence for the correlation between HIF-1α expression and the amount of stroma in colorectal cancers.
We added the result in the revised manuscript.
However, there was no significant correlation between HIF-1α expression and overall and recurrence-free survivals (p = 0.293 and p = 0.407, respectively).
Reference
Klaassen R, Steins A, Gurney-Champion OJ, Bijlsma MF, van Tienhoven G, Engelbrecht MRW, van Eijck CHJ, Suker M, Wilmink JW, Besselink MG, Busch OR, de Boer OJ, van de Vijver MJ, Hooijer GKJ, Verheij J, Stoker J, Nederveen AJ, van Laarhoven HWM. Pathological validation and prognostic potential of quantitative MRI in the characterization of pancreas cancer: preliminary experience. Mol Oncol. 2020;14(9):2176-2189.
Minor concern
Scale bars should be included in micrographs.
Response:
As a recommendation, we added the scale bar.

Reviewer 2 Report
Overall a well study was well designed and the correlation of MVD with stroma-high is novel. The correlation of stroma-high with worse OS validates what is already known and further supports the literature and that their patient cohorts are a good representation of the population.
One major comment I have is that I would like to see the OS curves with the MVD high and low patients. It is implied but not demonstrated that the MVD, although correlates with stroma-high, correlates with OS. The true relevance of their novel finding needs to be correlated with OS. This is lacking in the analysis and conclusion.
Furthermore another major comment is that they authors did not report on the interobserver agreement for TSR assessment. They need to have a minimum of two pathologist for the TSR evaluation in order to validate their findings. This is clearly described in the papers they reference relative to TSR.
Minor comments:
The authors report that HIF-1a of tumor cells was more highly expressed in the 27 stroma-high subgroup than in the stroma-low subgroup. They should discuss why stroma-high have higher levels of HIF-1a.
The sentence: “Stroma-low was significantly correlated with favorable tumor 32 behaviors and better survival”. Should be changed to “similar to the literature we observed …”.
In all previous reports mucinous were excluded as they have a unique pathology and prognosis.
The authors need to explain why they decided to included this as a subtype in their analysis.
The authors onluy aimed at the OS without analysing the prognostic value of the TSR in terms of disease-free survival (DFS), which is an essential outcome for CRC progression. (see ref. Dang et al UEG Nov 19, 2020)
The patient cohorts had not received neo-adjuvant therapy however the authors did not indicate if all the patients in the cohort received adjuvant therapy. This will affect the OS and PFS. This needs to be included in the methods section.
Author Response
We tried to address the points raised by the reviewers as best as we can.
Please see the attachment for more details.
Overall a well study was well designed and the correlation of MVD with stroma-high is novel. The correlation of stroma-high with worse OS validates what is already known and further supports the literature and that their patient cohorts are a good representation of the population.
One major comment I have is that I would like to see the OS curves with the MVD high and low patients. It is implied but not demonstrated that the MVD, although correlates with stroma-high, correlates with OS. The true relevance of their novel finding needs to be correlated with OS. This is lacking in the analysis and conclusion.
Response:
We checked the prognostic implication of MVD. However, there was no significant correlation between MVD and OS (p = 0.150). We added the results and figures in the revised manuscript.
Figure 4. Kaplan-Meier curves for overall survival (A) and recurrence-free survival (B) according to the microvessel density (MVD). Patients with high (green line) and low MVD (blue line) showed overall and recurrence-free survivals.
In addition, we added the comment for the prognostic implication of MVD as below:
In addition, there was no significant difference of overall and recurrence-free survivals between subgroups with high and low MVD (p = 0.150 and p = 0.178, respectively).
Furthermore, another major comment is that they authors did not report on the interobserver agreement for TSR assessment. They need to have a minimum of two pathologist for the TSR evaluation in order to validate their findings. This is clearly described in the papers they reference relative to TSR.
Response:
For conflicting cases, consensuses between three pathologists were reached in evaluating TSR as well as MVD. However, this comment for consensuses was only included in evaluating MVD, but not TSR. We added the comment in the revised manuscript as below:
Conflicting cases were re-examined, and consensuses between three pathologists (Kang, G., Pyo, J.S., and Kang, D.W.) were reached.
Minor comments:
The authors report that HIF-1a of tumor cells was more highly expressed in the stroma-high subgroup than in the stroma-low subgroup. They should discuss why stroma-high have higher levels of HIF-1a.
Response:
In pancreatic cancer, a stroma-high phenotype demonstrated high vessel density and high collagen fraction (Klaassen et al., 2020). From a biology perspective, these tumors are characterized by dense collagen content and good vascularization and are relatively well differentiated (Klaassen et al., 2020). Stroma-low tumors, where vessel density is low, would also be prone to develop hypoxia (Klaassen et al., 2020). However, it could not be found the evidence for the correlation between HIF-1α expression and the amount of stroma in colorectal cancers.
As a recommendation, we added the comments in the revised manuscript as below:
Pancreatic cancers with high stroma demonstrated high vessel density (Klaassen et al., 2020). Biologically, tumors with dense collagen content have good vascularization and are relatively well differentiated (Klaassen et al., 2020). Tumors with low stroma are associated with low vessel density and are involved with hypoxia (Klaassen et al., 2020). However, it could not be found the evidence for the correlation between HIF-1α expression and the amount of stroma in colorectal cancers.
Reference
Klaassen, R.; Steins, A.; Gurney-Champion, O.J.; Bijlsma, M.F.; van Tienhoven, G.; Engelbrecht, M.R.W.; van Eijck, C.H.J.; Suker, M.; Wilmink, J.W.; Besselink, M.G.; Busch, O.R.; de Boer, O.J.; van de Vijver, M.J.; Hooijer, G.K.J.; Verheij, J.; Stoker, J.; Nederveen, A.J.; van Laarhoven, H.W.M. Pathological validation and prognostic potential of quantitative MRI in the characterization of pancreas cancer: preliminary experience. Molecular oncology 2020, 14, 2176-2189, doi: 10.1002/1878-0261.12688.
The sentence: “Stroma-low was significantly correlated with favorable tumor behaviors and better survival”. Should be changed to “similar to the literature we observed …”.
Response:
We changed the comment as below:
Similar to the literature, we observed that stroma-low was significantly correlated with favorable tumor behaviors and better survival.
In all previous reports mucinous were excluded as they have a unique pathology and prognosis. The authors need to explain why they decided to included this as a subtype in their analysis.
Response:
In the previous study, van Pelt GW et al. (2018) described that mucus, but not a mucinous component, should be ignored for scoring. However, Huijbers et al. reported that mucinous component, but not mucin, included as the stromal area. If the entire tumor is composed of a mucinous component, the TSR is zero. This means that tumor component is no present in tumor with TSR 0%. Moreover, mucinous components are included mucin and floating tumor cells. The mucinous carcinoma defines as the carcinoma with greater than 50% mucinous component. Although the mucus without tumor cells is excluded, there was no specific reference for the interpretation of the mucinous component.
We described the detailed comments for evaluating mucinous component in the revised manuscript as below:
In the previous study, van Pelt GW et al. described that mucus, but not a mucinous component, should be ignored for scoring. However, Huijbers et al. reported that mucinous component, but not mucin, included as the stromal area. If the entire tumor is composed of a mucinous component, the TSR is zero. This means that tumor component is no present in tumor with TSR 0%. Moreover, mucinous components are included mucin and floating tumor cells. The mucinous carcinoma defines as the carcinoma with greater than 50% mucinous component. Although the mucus without tumor cells is excluded, there was no specific reference for the interpretation of the mucinous component.
Reference
van Pelt, G.W.; Kjær-Frifeldt, S.; van Krieken, J.H.J.M.; Al Dieri, R.; Morreau, H.; Tollenaar, R.A.E.M.; Sørensen, F.B.; Mesker, W.E. Scoring the tumor-stroma ratio in colon cancer: procedure and recommendations. Virchows Archiv 2018, 473, 405-412, doi: 10.1007/s00428-018-2408-z.
The authors only aimed at the OS without analyzing the prognostic value of the TSR in terms of disease-free survival (DFS), which is an essential outcome for CRC progression. (see ref. Dang et al UEG Nov 19, 2020)
Response:
As pointed out, we analyzed the prognostic values of TSR using overall and recurrence-free survivals.
Figure 3. Kaplan-Meier curves for overall survival (A) and recurrence-free survival (B) according to the tumor-stroma ratio (TSR). Patients with stroma-high (low-TSR: ≤50% of tumor cells relative to stroma; blue line) and stroma-low (high-TSR: >50% of tumor cells relative to stroma; green line) showed overall and recurrence-free survivals.
Recurrence-free survival (RFS) includes (1) any recurrence (local or regional, or distant) and (2) death due to any cause. Gourgou-Bourgade et al. described that recurrence-free survival is interchangeable with disease-free survival.
Reference
Gourgou-Bourgade, S.; Cameron, D.; Poortmans, P.; et al. Guidelines for time-to-event end point definitions in breast cancer trials: results of the DATECAN initiative (Definition for the Assessment of Time-to-event Endpoints in CANcer trials). Annals oncology. 2015, 873-9.
The patient cohorts had not received neo-adjuvant therapy however the authors did not indicate if all the patients in the cohort received adjuvant therapy. This will affect the OS and PFS. This needs to be included in the methods section.
Response:
Among 266 CRC cases, 2 cases finally had R1 resection. Remained 264 cases were R0 resection for the primary colorectal cancer. In our cohort, cases with CCRT were excluded. Unfortunately, it was difficult to verify the record for the adjuvant therapy.
We added the comment in the revised manuscript as below:
Cases with R0 and R1 resections were 264 and 2 cases, respectively.

Round 2
Reviewer 1 Report
The reviewer acknowledges a considerable revision of the manuscript, which has been made in line with the reviewers’ comments. The manuscript has been improved; however, some of the questions have not been properly responded.
Major concern
Representative pictures of both stroma-low and stroma-high cases should be presented.
Response:
We showed the representative pictures in Figure 1. Figure A and B represent colorectal cancers with stroma-low and stroma-high, respectively.
I meant the representative immunohistochemical pictures of HIF-1alpha and CD34 of both stroma-low and stroma-high cases should be presented. The pictures should represent the data shown in Table 3.
Minor concern
Scale bars should be included in micrographs.
Response:
As a recommendation, we added the scale bar.
Figure 2 lacks scale bars.
Author Response
The reviewer acknowledges a considerable revision of the manuscript, which has been made in line with the reviewers’ comments. The manuscript has been improved; however, some of the questions have not been properly responded.
Major concern
Representative pictures of both stroma-low and stroma-high cases should be presented.
Response:
We showed the representative pictures in Figure 1. Figure A and B represent colorectal cancers with stroma-low and stroma-high, respectively.
I meant the representative immunohistochemical pictures of HIF-1alpha and CD34 of both stroma-low and stroma-high cases should be presented. The pictures should represent the data shown in Table 3.
Response:
As a recommendation, we added the representative figures for the result of Table 3.
Figure 2. Representative immunohistochemical images for hypoxia-inducible factor-1α (HIF-1α) and CD34 (A-D). (A) Negative expression of HIF-1α in colorectal cancer with stroma-low (× 200). (B) Positive expression of HIF-1α in colorectal cancer with stroma-high (× 200). (C) Low CD34 expression of microvessel density (MVD) in colorectal cancer with stroma-low (× 200). (D) High CD34 expression of MVD in colorectal cancer with stroma-high (× 200). (Scale bar = 100 μm).
Minor concern
Scale bars should be included in micrographs.
Response:
As a recommendation, we added the scale bar.
Figure 2 lacks scale bars.
Response:
As a recommendation, we added the scale bar in Figure 2.

Reviewer 2 Report
Overall the authors addressed most of the comments and provided additional information to strengthen their findings. However, considering their additional analysis the authors need to further clarify the following: the MVD high was significantly higher in the Stroma high and low in the Stroma low, furthermore Stroma high correlated with poor OS however MVD high did not, as they indicated in the revised version. This is conflicting and not addressed in the paper. I would suggest they perhaps consider the % cutoff groups. As they indicated the % cutoff of TSR correlated with OS. For example 50% for TSR comparison of high and low demonstrated significant differences in OS. Is the same true for MVD? For example did they analyze the same cutoff % as they describe in the text for MVD? It could be that the MVD was significant in the 70% cutoff for TSR and this would explain why MVD did not correlate with survival. Unless I am missing something these needs to be clarified. This finding is relevant to understanding the biology of the cancer and the data and samples the authors have would be a good opportunity to further evaluate this finding. I believe this would be an important and unique finding for the authors.
Author Response
Overall, the authors addressed most of the comments and provided additional information to strengthen their findings. However, considering their additional analysis the authors need to further clarify the following: the MVD high was significantly higher in the Stroma high and low in the Stroma low, furthermore Stroma high correlated with poor OS however MVD high did not, as they indicated in the revised version. This is conflicting and not addressed in the paper. I would suggest they perhaps consider the % cutoff groups. As they indicated the % cutoff of TSR correlated with OS. For example, 50% for TSR comparison of high and low demonstrated significant differences in OS. Is the same true for MVD? For example, did they analyze the same cutoff % as they describe in the text for MVD? It could be that the MVD was significant in the 70% cutoff for TSR and this would explain why MVD did not correlate with survival. Unless I am missing something these needs to be clarified. This finding is relevant to understanding the biology of the cancer and the data and samples the authors have would be a good opportunity to further evaluate this finding. I believe this would be an important and unique finding for the authors.
Response:
As a recommendation, detailed analyses for the prognostic roles of MVD were performed. There was no significant correlation between MVD and OS in overall cases (as 1st round revision). Interestingly, in cases with ≥ stroma 70%, MVD was significantly correlated with overall and recurrence-free survivals (p = 0.004 and p = 0.005, respectively). However, in cases with < stroma 70%, MVD showed a significant correlation in recurrence-free survival, but not overall survival.
We added in the revised manuscript and supplementary Table (Table S1).
Additional subgroup analysis between high MVD and survivals according to the TSR was added in supplementary Table (Table S1).
|
Table S1. The correlation between high microvessel density and overall and recurrence-free survivals in various subgroups according to the tumor stroma ratio |
||
|
Subgroups |
Overall Survival (p-value) |
Recurrence-free Survival (p-value) |
|
≥ stroma 30% < stroma 30% |
0.132 0.458 |
0.297 0.458 |
|
≥ stroma 50% < stroma 50% |
0.210 0.866 |
0.339 0.820 |
|
≥ stroma 70% < stroma 70% |
0.004 0.053 |
0.005 0.042 |
p < 0.05 are highlighted in bold.

Round 3
Reviewer 2 Report
The authors addressed all my comments adequately.
This manuscript is a resubmission of an earlier submission. The following is a list of the peer review reports and author responses from that submission.
Round 1
Reviewer 1 Report
Abstract
It would be better to explicitly mention the different parts of the article, such as introduction, results and conclusion.
Introduction
- Is very short and somewhat short-sighted, and does not cover a lot of the article. More information on the potential (not fully understood but there are theories) of tumor microenvironment and information on the reason why the HIF-1 alpha and MVD are chosen.
- Moreover, prognostic impact of TSR in CRC has not been controversial, especially not in those articles referenced: those all state a clear prognostic impact.
- Why does it matter what the high-TSR rate of CRCs is? That is part of the patient characteristics table and not a goal for a study. That a subgroup analysis is performed is not information for the introduction but Material
Material and method
- Evaluation TSR: Incomplete, there are no references to why and what method is chosen, as well as it is unclear which area of the tumor they choose and why. Mucus is also included as a tumor, which is incorrect. If the specific term 'TSR' is used, it should be under the methods and descriptions under Mesker et al (2007), and the scoring method is described in detail in: van Pelt, G. W., Kjær-Frifeldt, S., van Krieken, J., Al Dieri, R., Morreau, H., Tollenaar, R., Sørensen, F. B., & Mesker, W. E. (2018). Scoring the tumor-stroma ratio in colon cancer: procedure and recommendations. Virchows Archiv : an international journal of pathology, 473(4), 405–412. (https://doi.org/10.1007/s00428-018-2408-z) This article needs to be read, used and cited at least before this article is reconsidered.
- TMA and IHC: It is unclear why MVD and HIF-1 alpha are chosen for assessment, since there is no explanation in the introduction. Also, it can be disputed why these are chosen, for the relationship is not (evident) clinically relevant.
- Statistics: Sentence 132-134 do not appear to be properly adjusted.
Results
- Patient characteristics: It is unclear why the patients with tumors stage I-II and III-IV are bundled together like the groups good-moderately differentiated tumors vs. poor Also, rectal tumors are included in the group with left sided colon carcinomas, but since these are an entirely different entity, you cannot group these together. In the literature these rectal tumors are thus often excluded. In addition, the used histology types are confusing. It is stated that there are mucinous tumors, tumors with a pure micropapillary component and those with a distribution. However, there is no mention of adenocarcinomas, signet ring cell carcinomas, neuroendocrine tumors, etc. Thus, it is not a research into the subtypes of histology what they claim.
- Here it is explained why they chose MVD and HIF-1 alpha, but this will have to be mentioned earlier.
- Authors explain in the part of "Correlation TSR and survival" that they looked at multiple cutoff values, while they did not mention this in Material at first (sentence 165-172). Also, again, look at Mesker et al (2007) and van Pelt et al (2018).
- Graph does not have clear explanation (the last sentence is not correct).
- Numbering of subchapters is not correct in the results heading: 3.1 - 3.3. - 3.3 and needs to be adjusted.
Discussion
- Sentence 185: "TSR defines the proportion of the tumor area in the overall tumor" is not correct, as will all articles by Mesker et al state.
- Sentence 188: "Tumors with rich stroma" is incorrect, but "Tumors rich in stroma" or "Tumors with a high percentage/amount of stroma" should be written.
- Authors indicate that there are several studies that give somewhat contradictory results in terms of prognosis, but the references given are incorrect: Vogelaar et al (2016) states actually that TSR is in fact of prognostic influence, and Scheer et al (in the end it turns out to be only Vogelaar et al and Scheer et al (2017) concludes that the TSR has potential as a prognostic factor (and this is moreover only in rectal cancers).
- Sentence 197-210: Here an extensive explanation of the different cutoff values / groups is given (and not in Material), but only the reference to Scheer et al (2017) is given and not even Mesker et al (2007), founder of the TSR, thus the last sentence '50% may be appropriate to distinguish' is incorrect, since this is ultimately already decided by Mesker et al (2007) and van Pelt et al (2018).
- Sentence 211-223: Here the article suddenly mentions automation while that is not previously mentioned anywhere and is not of added value. The sentences "selection of evaluation foci and of the whole slide can affect the value of TSR. Therefore, the evaluation criteria, rather than the evaluation method, may be necessary" (221-223) are incorrect: The first sentence is because they use a different TSR scoring without reference (whole slide) and last sentence is grammatically wrong in general (add: "... to determine.").
- Sentence 224: The histological subtypes of CRC are mentioned here, but is not all, and thus it cannot be claimed that all different histological subtypes are compared.
- Sentence 232: "Huijbers et al reported mucinous component was included as the stromal area" is incorrect; mucus is excluded in the scoring of TSR as van Pelt et al (2018) informs in detail.
- Sentence 244: "Mesker et al reported that TSR was useful in predicting prognosis for stage I-III colon cancers", not T4 because of the "predicting prognosis of TSR being limited in locally advanced tumors", is not correct.
- Sentence 259-262: Here the relationship between rapid tumor growth and hypoxia is explained but not a clinical relevance.
Conclusion
- Sentence 266 is said that "a 50% cutoff may be appropriate", while it has already been determined by Mesker et al (2007), and was not an objective of the study.
General
- Using the terms 'high-TSR' and 'low-TSR' is confusing, perferred terms are 'stroma-high' and 'stroma-low'.
- It is also unclear why HIF-1 alpha and MVD are chosen, whether these could have a causal relationship or whether that is even logical in all and irrelevant (e.g. stroma simply contains more vessels and what relevance does the hypoxia have for prognosis?).
Reviewer 2 Report
1. The authors should assess and describe the intratumoral heterogeneity of the TSR in the tumor. "TSR defines the proportion of the tumor area in the overall tumor. " is insufficient.
2. The authors didn't show the immunohistochemical or captured and converted images of HIF-1α and MVD. Mechanisms of how TSR correlates with MVD or HIF-1α expression have not been revealed. This is not acceptable.
Round 2
Reviewer 2 Report
I asked the authors to assess the intratumoral heterogeneity of TSR and they added the comments in the Materials and Methods section. The authors argue that they have chosen the most representative cross-section to evaluate TSR. This is far too subjective. At least two surgical pathologists must measure the TSR independently without the clinical and pathological parameters or outcomes of the patients. Using image processing software can also be accepted.
I also asked the authors to reveal the cause of the positive correlations between TSR and MVD or hypoxic status. This point is at all improved. The data shown in this paper is based on the phenomenology analysis and the authors should clarify the cause of the results.
For the reasons mentioned above, this paper can not be accepted in this journal. However, the results are valuable in part, I recommend the authors to apply to other journals.